# Video Head Impulse Test (vHIT): Value of Gain and Refixation Saccades in Unilateral Vestibular Neuritis

**DOI:** 10.3390/jcm11123467

**Published:** 2022-06-16

**Authors:** George Psillas, Ioanna Petrou, Athanasia Printza, Ioanna Sfakianaki, Paris Binos, Sofia Anastasiadou, Jiannis Constantinidis

**Affiliations:** 11st Otolaryngology Department, School of Medicine, Faculty of Health Sciences, Aristotle University of Thessaloniki, AHEPA Hospital, Stilponos Kyriakidi St., 546 36 Thessaloniki, Greece; petrouio@hotmail.gr (I.P.); nan@med.auth.gr (A.P.); g_sfakianaki@yahoo.com (I.S.); anastasiadousof23@gmail.com (S.A.); janconst@otenet.gr (J.C.); 2Department of Rehabilitation Sciences, Cyprus University of Technology, Limassol 3036, Cyprus; paris.binos@cut.ac.cy

**Keywords:** video head impulse test, saccades, vestibular gain, vestibular neuritis, vestibuloocular reflex, semicircular canal

## Abstract

The aim of this study was to evaluate gain and refixation saccades (covert and overt) using a video head impulse test (vHIT) in the horizontal and vertical planes in patients after the onset of unilateral acute vestibular neuritis (AVN). Thirty-five patients were examined in the acute stage of AVN and at follow-up (range, 6–30 months); a control group of 32 healthy subjects also participated. At onset, the mean gain was significantly lower on the affected side in all of the semi-circular canal planes, mainly in the horizontal canal plane, and saccades (covert and overt) were more prevalent in the horizontal compared to the vertical canal planes. Multi-canal affection occurred more frequently (80% for gain, 71% for saccades) than isolated canal affection. At follow-up, which ranged from 6 to 30 months, the gain was recovered in all of the canals (anterior in 50%, horizontal in 42.8%, and posterior canal in 41.1% of cases), while covert and overt saccades were reduced in the horizontal and vertical planes. However, covert saccades were still recorded in a greater proportion (69%) than overt saccades (57%) in the horizontal plane and at a lower rate in the vertical planes. The compensatory mechanisms after AVN mainly involve the horizontal canal, as the refixation saccades—especially covert ones—were more frequently recorded in the horizontal than vertical canals.

## 1. Introduction

Acute vestibular neuritis (AVN) is a clinical condition characterized by acute, prolonged vertigo of peripheral origin with nausea and vomiting followed by a period of unsteadiness [1]. AVN accounts for 3.2–9% of patients visiting a dizziness center and has an incidence of 3.5 per 100,000 population [2]. The leading theory for the etiology of AVN is viral inflammation processes, with the herpes simplex virus being most commonly implicated [3].

AVN involves both the superior and inferior vestibular nerves and may affect them separately. The superior vestibular nerve innervates the horizontal or lateral semi-circular canal (HC), the anterior or superior canal (AC), and the utricle of the labyrinth, while the inferior vestibular nerve innervates the posterior or inferior canal (PC) and the saccule.

The vestibulo–ocular reflex (VOR) is an important reflex that aims to ensure gaze stability to provide stable vision during rapid head rotations [4]. This is assessed by asking a patient to maintain their gaze on a target while an operator applies unpredictable and abrupt but small head turns in the horizontal and vertical planes [5,6]. If the VOR function is intact, the subject maintains fixation on the target during head rotations by generating eye movements that are equal and opposite to the rotation of the head; however, if their VOR function is deficient (i.e., during AVN with unilateral reduced horizontal or vertical canal function), the eyes of the patient can fail to fix on the target, and the patient must make corrective refixation saccades (eye movements) back to the target. The saccades that can be detected by the operator following the end of the head movements are known as overt saccades, while those that occur during the head impulses are known as covert saccades [4,5,6]. Covert saccades are invisible to the clinician; fortunately, the video head impulse test (vHIT) can recognize covert saccades and measure the VOR gain [7]. The VOR gain is the ratio of the size of the slow-phase eye movement response to the size of the passive head movement stimulus, which is decreased in peripheral vestibular disorders [8]. Thus, the vHIT allows for objective analysis of VOR gain, VOR gain asymmetry (analogous to caloric canal paresis), and refixation saccades of all six semi-circular canals [9].

In this study, we aim to prospectively evaluate the VOR gain and refixation saccades (covert and overt) in AVN from the acute stage to the follow-up (ranging from 6 to 30 months). Although previous studies have assessed, through the use of vHIT, the VOR gain in the horizontal and vertical semi-circular canals in patients suffering from AVN [10,11,12,13], there has been no study analyzing the VOR gain and refixation saccades for AVN in both the horizontal and vertical planes.

## 2. Materials and Methods

Thirty-five patients, including 28 men and 7 women (age range, 24–77 years; mean age, 52.25; SD, ±13.23 years), participated in our study. The patients were admitted consecutively after diagnosed with acute vestibular neuritis (AVN), according to the following criteria [1,10]: (a) sudden rotatory vertigo lasting more than 24 h (accompanied by nausea and vomiting), (b) unidirectional horizontal-torsional spontaneous nystagmus towards the unaffected side, (c) a positive Unterberger test, where the patient was asked to undertake stationary stepping for one minute with their eyes closed and a positive test was indicated by rotational movement of the patient towards the side of the lesion, (d) unilateral affection, and (e) normal hearing.

The exclusion criteria were age under 18 years old, pregnancy, auditory or neurological disorders, history of psychiatric disorder, patients suffering from diabetes, patients with neck stiffness, a history of taking CNS depressants, excessive blinking, and a history of recent eye surgery. All patients were treated with intravenous corticosteroids during their stay at the hospital and, on discharge, they performed a vestibular rehabilitation program for 3 weeks. We had a long-term follow-up period (mean, 14.37; SD, ±7.69 months) that ranged from 6 to 30 months.

In addition, all of patients were required to complete the Greek version of the Dizziness Handicap Inventory (DHI) questionnaire (25 questions) during the acute phase and at the follow-up examination. For each item, the following scores were assigned: no = 0, sometimes = 2, yes = 4. A DHI score of 16–35 points indicates mild disability, 36–53 points indicates moderate disability, and more than 54 points indicates severe disability [14]. Pre- and post-treatment would have to differ by at least 18 points (95% CI for a true change) before the interaction could be said to have effected a significant change in a self-perceived handicap [15].

### 2.1. Control Group

The control group consisted of 32 healthy adults, including 13 men and 19 women (age range, 22–74 years; mean age, 45.18; SD, ±15.15 years), who were evenly distributed in all age groups. None of them had any history of vestibular, auditory, and/or neurological disorders.

Written informed consent was obtained from all subjects participating in this study. This study was approved by the ethics committee of the Aristotle University of Thessaloniki.

### 2.2. vHIT

vHIT was performed on all subjects using an EyeSeeCam system (Interacoustics, Middelfart, Denmark) at the time of the initial presentation, within 10 days of the onset of symptoms, and was repeated at the follow-up visit. All six semi-circular canals were evaluated. The system consists of a pair of lightweight goggles with a high-speed camera for recording eye movements and a sensor that measures head velocity. Participants were instructed to fixate on a target on the wall 1.2 m away, while the examiner stood behind them and performed ten unpredictable, low amplitude (10–20°) head impulses for all three canal planes (horizontal, right anterior/left posterior, left anterior/right posterior) and for each side; in general, the horizontal head velocities were higher (150–250 °/s) than the vertical head velocities (50–150 °/s). For the horizontal plane, the examiner placed their hands on the patient’s jaw while, for the vertical plane, the dominant hand was placed on the top of the head and the other under the chin. For the right/left horizontal canal plane, horizontal head turns were performed to each side, always starting from the center (Figure 1). For the right anterior/left posterior canal testing, the subject turned 45° to the left and the impulses were delivered by either rotating the head forward or backward (always asking the patient to fixate on the target) while, for the left anterior/right posterior canal testing, the subject turned 45° to the right and the impulses were delivered by rotating the head either forward or backward.

Each examined plane was described as: (a) Affected HC plane as affected HC/contralateral unaffected HC plane, (b) affected AC plane as affected AC/contralateral unaffected PC plane, and (c) affected PC plane as affected PC/contralateral unaffected AC plane (Figure 1).

The system’s software calculates the VOR gain as the ratio of eye to head peak velocity from the average of head impulses and gain asymmetry (%) between the affected and unaffected canals for each plane, according to the following equation: GA = ((Gc − Gi)/(Gc + Gi)) × 100%, where GA: gain asymmetry, Gi: gain for ipsilateral canal and Gc: gain for contralateral canal. The EyeSeeCam system uses the regression method for VOR gain and gain asymmetry, which allows for graphical data analysis over the entire velocity range of head impulses; following the test completion, it provides the average regression plot slope (a best-fit line through data points at different head velocities with accompanying gain values). The instantaneous median VOR at 60 ms was used for measuring the horizontal gain. A perfect VOR gain is a gain of 1.0 (eye movement is exactly equal and opposite to head movements); while a VOR gain lower than 0.80 has been proposed as cut-point between normal and abnormally low VOR gain [16]. The gains measured in each plane are referred to as HC gain, AC gain, and PC gain, respectively. Depending on which ear was involved, the gain of the affected and the unaffected side was compared to the gain of the right and left ear of the healthy subjects (control group), respectively.

Pathological saccades were considered in a patient when their peak velocity was >100 °/s and their repeatability was observed in more than 50% of total impulses for each plane. The presence of pathological saccades in each plane was described as HC overt or covert saccades, AC overt or covert saccades, and PC overt or covert saccades, accordingly (Figure 2). In our study, the frequency of pathological saccades was counted in each patient for all the canals and the incidence of patients with pathological saccades was also calculated.

## 3. Statistical Analysis

Continuous variables are reported as mean and standard deviation, after being checked for regularity with the Kolmogorov–Smirnov test. Categorical variables were reported to as frequencies and relative frequencies (%). Student’s *t*-test (for two categories) or one-way ANOVA (for three or more categories) were used to compare the mean values of the continuous variables, with respect to the categories of a qualitative variable. Bonferroni correction was used for post hoc analysis. Pearson’s correlation coefficient was also used to investigate the correlations between continuous variables. In all cases, we considered the results to have statistical significance when the *p*-values were less than 0.05. The SPSS v.25.0 software (IBM Corp., Armonk, NY, USA) was used for statistical analysis.

## 4. Results

In the 35 patients with AVN, at the acute stage, the mean gain was significantly lower (compared to control group) on the affected side in all of the canal planes, but mainly in the horizontal canal plane (Table 1). In the HC plane, the gain was abnormal in 28 (80%) out of 35 patients while, at the follow-up, the gain was normal in 12 (42.8%) out of 28 patients. In the AC plane, the gain was also abnormal in 28 (80%) out of 35 patients while, at the follow-up, the gain was normal in a greater proportion (50%, 14/28). In the PC plane, the gain was abnormal at a lower rate, compared to the other canals (48.5%, 17/35) while, at the follow-up, the gain was normal in 7 (41.1%) out of 17 patients. Interestingly, on the unaffected side, the mean gain at acute stage was found to be smaller than that of the control group in all the canal planes (Table 1). Moreover, according to vHIT measurements on the affected side, the gain improved at the follow-up in all the canal planes, but significantly in the HC (*p* < 0.01) and AC (*p* < 0.001) canal planes, respectively (Table 1). On the affected side, the mean gain asymmetry was decreased at the follow-up in all of the canal planes, but was more significant in the affected AC/unaffected PC planes (Table 2).

Initially, the incidence of pathological covert saccades in all planes (94% in the HC, 57% in the AC, and 31% in the PC plane) was somewhat higher than that of overt saccades (86%, 49% and 26%, respectively); see Table 3. At the follow-up examination, the occurrence of covert saccades was significantly decreased in the HC (94% vs. 69%; *p* < 0.05) and AC planes (57% vs. 17%; *p* < 0.01). Similarly, the overt saccades were also significantly reduced in the HC (86% vs. 57%; *p* < 0.05) and AC planes (49% vs. 11%; *p* < 0.01). At the follow-up, regarding the total rate of covert and/or overt saccades for each canal plane, a relatively high incidence of saccades (90%) was found in the HC, compared to in the other canal planes (Table 3).

At the acute stage, based on either gain values or incidence of saccades, the number of patients with multi-canal affection was greater than that with isolated canal affection (Table 4). More specifically, in terms of gain, 80% (28/35) showed multi-canal affection vs. 14% (5/35) for isolated canal affection. In terms of saccades (overt and covert), 71% (25/35) showed multi-canal affection vs. 28.5% (10/35) for isolated canal affection. Regarding gain values, the HC was involved in 28 patients, the AC in 28 patients, and the PC in 17 patients; notably, in two patients suffering from AVN, the gain was normal in all the canals. Refixation saccades were registered in the HC plane of 34 patients, in the AC plane of 24 patients, and in the PC plane of 13 patients (Table 4).

According to the DHI (acute stage), 5 (14.2%) patients had mild disability, 12 (34.2%) had moderate disability, and 18 (51.4%) patients had severe disability; the mean total DHI was 52.17 ± 13.23. At the follow-up, almost all the patients presented a normal DHI score, except for 5 (14.2%) patients who had mild disability; the mean total DHI was 7.94 ± 9.52. In the initial stage, the gain values and the gain asymmetry of each semi-circular canal on the affected and unaffected sides were not statistically correlated with the DHI measurements. At the follow-up, no statistical correlations were performed, as all of the DHI scores were below 35 and no clinical significance was expected.

## 5. Discussion

According to our results, the majority of patients (80%) with AVN had lower VOR gain in the HC and AC planes, compared to the PC plane (48.5%). The cause of this high affection of both HC and AC planes could be attributed to anatomical particularities of the bony channel of the superior vestibular nerve, as it is seven times longer, narrower, and has much more spicules along its course, compared to that of the inferior vestibular nerve; this makes the superior vestibular nerve more susceptible to compressive injury and ischemia [17].

In the period after AVN, a progressive restoration of the semi-circular canal gain was observed in our patients, mainly regarding the HC and AC gain (Table 1). A central compensation process, based on synaptic and neuronal plasticity, has been reported to be responsible for the gain improvement across all the canal planes [18]. Central mechanisms have also been implicated in the upregulation of the unaffected side [19]; at the acute stage, our lower gain values were effectively compensated as, otherwise, the lack of disinhibitory input through commissural pathways would cause a greater gain reduction at higher accelerations on the unaffected side [19]. The gain asymmetry decreased on the affected side, mainly in the affected AC/unaffected PC plane (Table 2). However, Allum and Honegger [20] reported that the mean gain asymmetry change between onset and 5 weeks was greater (*p* < 0.001) in the horizontal canal plane (from 36.9% to 19.4%) than in the other planes.

Although it has been reported that 3–6 months are necessary to observe a stabilization of the gain [21], a time of more than one year and a half may be required for higher horizontal head velocities [22]. In our study, our follow-up period was extended up to 30 months, and we found that the gain was best recovered in the affected AC plane (50%) compared to in the affected HC (42.8%) and PC (41.1%) planes. Büki et al. [23] showed that normal HC gain occurred in 55%, AC gain in 38%, and PC gain in 38% of 40 patients, supporting the idea that vertical canals recover less effectively. Furthermore, Magliulo et al. [13] reported that the PC gain was better normal in 42.1% (8/19), HC gain in 37.1% (13/35), and AC gain in 25.8% (8/31). The discrepancy between these findings could be attributed to the short-term follow-up of the two abovementioned studies: two months [23] and three months [13], respectively.

Despite the fact that the VOR gain was not fully recovered in all cases, the majority of patients returned to their daily living activities. The question was how, after AVN, can inadequate recovery of slow phase eye velocity compensate for the VOR deficit to natural head accelerations. Previous studies [8,24] have suggested that the presence of refixation saccades may play a very important role in stabilizing the gaze in space during head rotations. In patients with unilateral vestibular loss, these saccades may act to minimize the effect of the dynamic VOR deficit, and could also be considered as part of a compensatory mechanism to overcome the loss of peripheral vestibular function [4]. In our study, at the acute stage, the saccades were more obviously detected in the HC plane (covert 94%, overt 86%, mixed 97%; Table 3), compared to in vertical canal planes. It has been demonstrated [25] that averaged VOR speed gains were greater in the HC plane, compared to those of vertical canals, in normal subjects; therefore, it is possible that the saccade recordings are better produced during head rotational accelerations on the horizontal plane.

As mentioned previously, no other report in the literature has assessed the refixation saccades for AVN in the vertical planes. According to our results (Table 4), at the follow-up, both covert and overt saccades—especially those on the affected AC plane—were significantly reduced in all of the planes. It is worth noting that, in long-term re-examination, HC covert saccades were recorded at a greater proportion in our patients, compared to HC overt saccades (69% vs. 57%). Fu et al. [5] also registered, six months after AVN 87% covert vs. 59% overt saccades. It has been reported [6] that, as vestibular compensation is promoted, there is an evolution in the saccade pattern from overt to covert saccades. However, a higher percentage of overt compared to covert frequency has been observed 15 days after AVN (54.6% overt vs. 13.3% covert) [4], 87% overt vs. 43% covert at 1 month after AVN [26], and 73.4% overt vs. 14.7% covert at 4 months after AVN [27]. The role of covert and overt saccades after AVN has not yet been clarified. It has been proposed that covert saccades may contribute to the compensation of inadequate VOR response and to symptom-associated recovery [5]. Conversely, a high prevalence of overt saccades has been related to a worse prognosis after acute AVN [28]. Wettstein et al. [29] reported that patients suffering from AVN and higher percentage of covert saccades had better dynamic visual acuity; on the contrary, patients with few covert saccades and/or high amplitude of cumulative overt saccades had poor dynamic visual acuity. Navari et al. [27] supported the idea that residual disability after acute unilateral vestibulopathy was related with lower high-velocity VOR gain, increased gain asymmetry, and increased overt saccades number and amplitude. It seems that the presence of covert saccades is more essential for the vestibular compensation process and would faster facilitate the recovery, taking into account that almost all of our patients had a favorable outcome after AVN.

Due to the clinical application of the vHIT, the interpretation of the results was based on the gain value, which was used to determine whether the vestibular function could be classified as either normal or pathological. It has also been reported that it is advisable to check both gain and refixation saccades when performing vHIT to detect vestibular hypofunction [26]. However, patients suffering from vestibular disorders in which the gain was normal on vHIT testing have been reported, where only the presence of refixation saccades indicated the existence of peripheral vestibulopathy [30]. Thus, the occurrence of saccades could be more reliable than the gain value for the evaluation of vestibular function [31]. Two of our patients with AVN showed normal gain in all canals, raising a question regarding the pathophysiology of this disease; however, both of these patients showed refixation saccades (Table 4). Unlike the gain results (Table 4), and based on the saccade findings, there was no AC- or PC-isolated AVN, but more HC-isolated AVN (10 cases).

According to our results, multi-canal affection occurred more frequently (80% for gain, 71% for saccades) than isolated canal affection (see Table 4). Thus, at least two canals were usually implicated in the pathogenesis of AVN. We have retrieved reports that assessed the semi-circular canal involvement (horizontal and verticals) through vHIT in patients suffering from AVN, and found five studies measuring the gain in these canals (but not the presence of saccades). Multi-canal affection was reported in 60% [11], 62.5% [27], 68.9% [12], 75% [10], and 80% of cases [13], respectively; in almost all of these studies, both the HC and AC gain, or the gain of all three canals (HC + AC + PC), were more frequently found to be abnormal. However, only sporadic cases with isolated horizontal or anterior canal AVN have been described in the literature [10,23]. It seems that, in most AVN cases, the damage is extensive rather than selective; as, for example, both superior and inferior vestibular nerve AVN is more frequent than superior or inferior vestibular nerve AVN alone [10,13,27].

The DHI is the most widely used self-reported measurement of patients with dizziness [32]. Our long-term mean total DHI after AVN was 7.94, indicating a complete recovery. Other studies assessing long-term DHI results after AVN have presented comparable values, such as 12.51 (follow-up: 5.3 y) [33] and 18.3 (follow-up: 3 y) [11]. In our study, at the acute stage, no statistically significant correlation between the vHIT parameters and the DHI values was found. Similarly, Riska et al. [4] also did not find a significant difference in DHI score as a function of saccade group (DHI score: 48 for covert, 47 for overt, 54 for mixed saccades) less than one month after AVN. Patel et al. [11] reported that 10% of their patients, 3–36 months after AVN, showed DHI scores more than 35, but they did not find a correlation between ipsilesional high velocity VOR gain or gain asymmetry of the single or combined HC, AC, and PC with the DHI scores. However, Manzari et al. [6] observed a statistical correlation between the HC gain and DHI values six weeks after AVN. Fu et al. [5] compared two groups of patients at 6 months after UVN—28 patients without or with mild disability (according to DHI) to 19 patients with moderate to severe disability—and reported that the proportion of patients with presence of covert saccades in the first group was significantly higher than that in the latter group.

## 6. Conclusions

In the relatively long follow-up period after AVN, the vHIT gain was found to be increased on the affected side in all of the three semi-circular canals, mainly in those innervated by the superior vestibular nerve (i.e., HC and AC canals). The vHIT gain was recovered in all canals, and both covert and overt saccades were reduced in all the canal planes—notably, they were more significantly in the affected AC/unaffected PC plane. The compensatory mechanisms after AVN mainly involved the HC, as the refixation saccades—especially the covert compared to overt saccades—were more frequently detected in the HC plane than in the vertical canal planes. In AVN, multi-canal affection occurred more frequently than isolated semi-circular canal affection, indicating that the vestibular nerve damage is usually more extensive than selective.

## Figures and Tables

**Figure 1 jcm-11-03467-f001:**
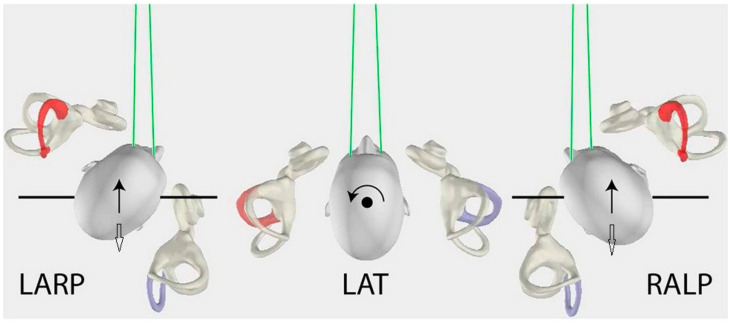
In the middle, affected HC (horizontal canal) plane, the head movements (circle black arrow) for left affected (red) HC/contralateral unaffected right (purple) HC, as viewed from above. On the left, affected AC (anterior canal) plane, the head movements (black arrow) for left affected (red) AC/contralateral unaffected right (purple) posterior canal (PC). On the right, affected PC plane, the head movements (white arrow) for left affected (purple) PC/contralateral unaffected right (red) AC. Inversely, on the left, affected PC plane, the head movements (white arrow) for right affected (purple) PC/contralateral unaffected left (red) AC. On the right, affected AC plane, the head movements (black arrow) for right affected (red) AC/contralateral unaffected left (purple) PC (with the permission of Curthoys IS). LARP: left anterior & right posterior canal, LAT: lateral or horizontal canals, RALP: right anterior & left posterior canal.

**Figure 2 jcm-11-03467-f002:**
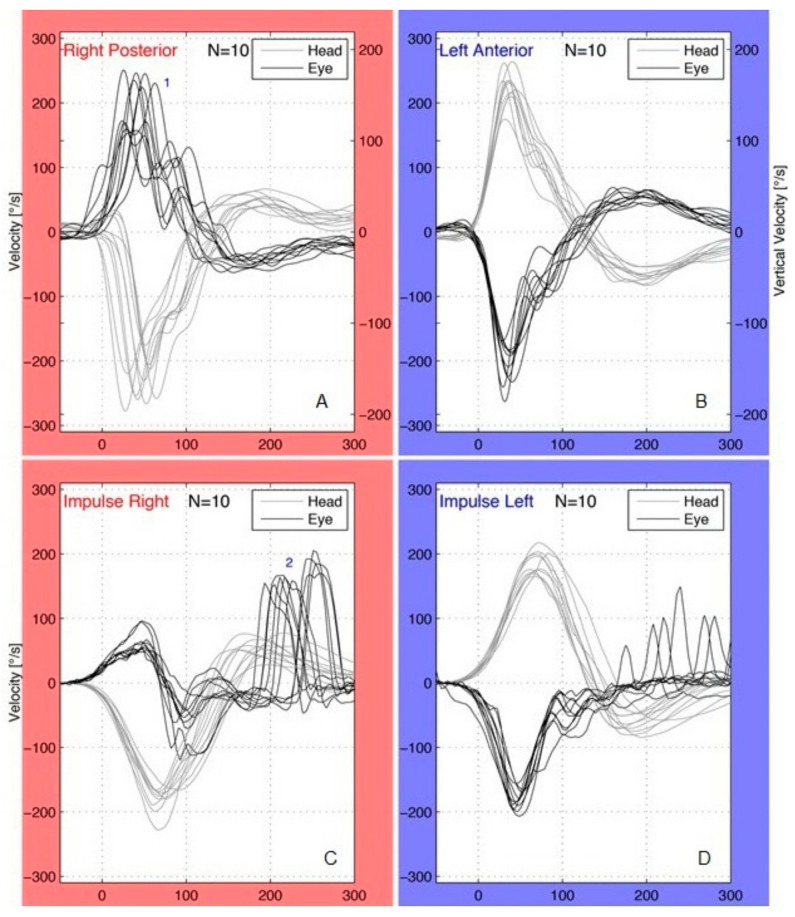
Patient with right acute vestibular neuritis (**A**,**B**) and affected posterior canal (PC) plane: pathological covert saccades (1) were present in the right PC (**A**). Covert saccades occurred during the head impulses and their peak velocity was more than 100 °/s; no saccades were observed in the left unaffected anterior canal (**B**). Patient with right acute vestibular neuritis (**C**,**D**) and affected horizontal canal (HC) plane: pathological overt saccades (2) were present in the right HC (**C**) following the end of the head movements; the saccades found in the left HC (**D**) were not pathological (peak velocity <100 °/s, repeatability <50% of total impulses).

**Table 1 jcm-11-03467-t001:** Gain values of vHIT in vestibular neuritis at the acute stage and follow-up during head movements towards the affected and unaffected side, respectively, in horizontal (HC), posterior (PC) and anterior (AC) plane (see also Figure 1). *** *p* < 0.001, ** *p* < 0.01, * *p* < 0.05. SD: standard deviation, C.I.: confidence interval.

Gain		Mean ± SD	Significance	95% C.I. for Mean
Affected HC	Acute stage	0.54 ± 0.36	Acute—control *** Acute—follow-up **	0.41–0.67
	Follow-up	0.77 ± 0.31		0.66–0.88
	Control	0.86 ± 0.19		0.80–0.93
Unaffected HC	Acute stage	0.84 ± 0.36	Acute—control **	0.72–0.97
	Follow-up	0.98 ± 0.32		0.86–1.09
	Control	1.10 ± 0.28		1.00–1.20
Affected AC	Acute stage	0.64 ± 0.16	Acute—control * Acute—follow-up ***	0.62–0.74
	Follow-up	0.83 ± 0.15		0.77–0.88
	Control	0.78 ± 0.10		0.74–0.82
Unaffected AC	Acute stage	0.79 ± 0.16	Acute—control *	0.73–0.85
	Follow-up	0.84 ± 0.18		0.77–0.90
	Control	0.90 ± 0.10		0.86–0.94
Affected PC	Acute stage	0.78 ± 0.17	Acute—control *	0.72–0.84
	Follow-up	0.83 ± 0.16		0.77–0.89
	Control	0.87 ± 0.12		0.83–0.92
Unaffected PC	Acute stage	0.77 ± 0.15	Acute—control *	0.72–0.83
	Follow-up	0.85 ± 0.14		0.80–0.90
	Control	0.86 ± 0.10		0.82–0.90

**Table 2 jcm-11-03467-t002:** Gain asymmetry values of vHIT in vestibular neuritis at the acute stage and follow-up in the affected horizontal canal (HC), posterior canal (PC) and anterior canal (HC) plane. * *p* < 0.05.

Gain Asymmetry		Mean ± SD	Significance	95% C.I. for Mean
Affected HC	Acute stage	17.86 ± 28.50		8.07–27.65
	Follow-up	9.83 ± 21.10		2.58–17.08
	Control	12.81 ± 10.78		8.92–16.7
Affected AC	Acute stage	7.80 ± 11.16	Acute—Follow-up *	3.97–11.63
	Follow-up	1.60 ± 8.45		−1.30–4.50
	Control	5.97 ± 6.31		3.69–8.24
Affected PC	Acute stage	0.46 ± 11.42	Follow-up—control *	−3.47–4.38
	Follow-up	0.11 ± 11.13		−3.71–3.94
	Control	5.94 ± 5.32		4.02–7.86

**Table 3 jcm-11-03467-t003:** Incidence of pathological covert and overt saccades during vHIT in patients suffering from vestibular neuritis at the acute stage and follow-up affected side. HC: horizontal canal plane, AC: anterior canal plane, PC: posterior canal plane. *** *p* < 0.001, ** *p* < 0.01, * *p* < 0.05.

Covert						Overt					Total (Covert and/or Overt Saccades)				
Affected Side	Acute Stage		Follow-Up			Acute Stage		Follow-Up			Acute Stage		Follow-Up		
	N	%	N	%	*p*-Value	N	%	N	%	*p*-Value	N	%	N	%	*p*-Value
HC	33	94%	24	69%	*	30	86%	20	57%	*	34	97%	28	90%	
AC	20	57%	6	17%	**	17	49%	4	11%	**	24	69%	8	23%	***
PC	11	31%	5	14%		9	26%	5	14%		13	37%	6	17%	

**Table 4 jcm-11-03467-t004:** Isolated and multi-canal involvement according to vHIT parameters (gain, covert saccades, and overt saccades) at the acute stage of vestibular neuritis (affected side). HC: horizontal canal plane, AC: anterior canal plane, PC: posterior canal plane.

Isolated Canal Involvement						Multi-Canal Involvement				
	HC	AC	PC		HC + AC	HC + PC	AC + PC	HC + AC + PC	Normal	Total
Gain	2	2	1		12	2	2	12	2	35
Saccades	10	-	-		12	1	1	11	-	35

## Data Availability

The data presented in this study are available upon request from the corresponding author.

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
