# Peer review of "Video Head Impulse Test (vHIT): Value of Gain and Refixation Saccades in Unilateral Vestibular Neuritis"

_jcm, 2022, doi:10.3390/jcm11123467_

Round 1

Reviewer 1 Report

Review: Psillas et al. – Video Head Impulse Test (vHIT)…..

The paper presents a lot of data with a seemingly exhaustive statistical analysis. There are few complaints except:

1)    How was head acceleration measured? Did any of the variances (150-250 deg/sec2) affect the outcome. Also: semicircular canals detect accelerations, so make sure you use the right dimension, i.e., “deg/sec2”.

2)    Please, when mentioning the head impulse test first give credit to Halmagyi who described it first (Curthois-Halmagyi head jerk).

3)    The English is understandable and acceptable, but it reads a little clumsy at times. I suggest language editing for clarity and interpunctation.

Minor:

l. 16: replace: ”detected” with “prevalent”

l. 17: “..vertical canal planes”.

l. 18: “…than isolated canal affections.”

l. 20: remove “the”

l. 21: “…recorded at greater proportions (69%) than covert…”

l. 23: remove the two “the”.

l. 67: “Unterberger”

l. 71: “i.v.”

l. 75: “…were required to complete the Greek…”

l. 85: “….by the ethics committee of Aristotle…”

l. 87: “…was performed on all…”

ll. 130, 132, 134 and elsewhere: replace “normalized” with “normal”

l. 202: remove “ ‘ study”

l. 203: “normal” (not “normalized”)

l. 223: “…for AVN in vertical…”

l. 229: “saccade pattern”

l. 231: “shown”

l. 236: “reported” (not “supported”)

ll. 238/239: the half-sentence staring “patients….” needs to be reworked. It should end with “…had poor dynamic visual acuity”. The tail “..in patients with unilateral peripheral vestibular loss” makes no sense.

l. 241: “..increased gain asymmetry..”

l. 243: remove “that”

l. 269: “…AVN alone [9, 12, 25]”

Author Response

We thank the reviewer for his criticisms in terms of improving our article.

  1. How was head acceleration measured? Did any of the variances (150-250 deg/sec2) affect the outcome. Also: semicircular canals detect accelerations, so make sure you use the right dimension, i.e., “deg/sec2”.

We have performed our tests based on the company standardizations (EyeSeeCam system, Interacoustics, Denmark) consisting in head impulses of small amplitude (5-15o/sec) and consistent peak velocity, which was relatively high (150-250o/sec) and no acceleration. After the completion of the test the company software calculates the gain value, which is defined as the eye velocity divided by the head velocity. Although the normal head movements have very high accelerations (4,000/s2) (Grossman GE, 1989), the device has been scheduled to calculate on degrees/sec (not on degrees/sec2), in order to estimate eye and head velocities, respectively (https://www.interacoustics.com/balance/software/eyeseecam). The EyeSeeCam system uses the regression gain method, which allows for graphical data analysis over the entire velocity range, showing the total of head impulses and not selectively every velocity measurement; following the test completion, it provides the average regression plot slope (a best-fit line through data points at different head velocities with accompanying gain values) (Hougaard DD and Abrahamsen ER, Journal of Visualized Experiments, 2019;146:e59012, pg 1-14). Thus, the system did not allow us to assess separately the variances across the head velocities, although they were illustrated on the screen.

  1. Please, when mentioning the head impulse test first give credit to Halmagyi who described it first (Curthois-Halmagyi head jerk).

As you suggested, we have added the first report for vHIT of Curthoys and Halmagyi in the references.

  1. The English is understandable and acceptable, but it reads a little clumsy at times. I suggest language editing for clarity and interpunctation.

We revised our English wording and had our manuscript polished by English-language editing service.

Thank you very much for taking the time to comment, promoting our expression more clearly and getting improvement.

Best regards

Reviewer 2 Report

This study present information on the presentation of loss of VOR gain and corrective saccades after Acute vestibular neuritis using the vHIT technology.  

Additional information on covert and overt saccadic presentation in all three canal planes adds to the literature.  Below are line by line comments and recommendations that would improve the manuscript. 

In Abstract  Page 1 line 18 Time for follow should be clearly stated. 

Introduction:  page 2 line 50.  Size of the “slow phase eye movement response.”  Eliminate term corrective.  That would be associated with the corrective saccades not the VOR.

Materials=Methods

Line 63-64.  I would change location of word consecutively.  “The patients were admitted consecutively after diagnosed with AVN (no need to respell out) according to the following criteria:….”

Line 67:  Please reference the Uterberger test or describe.   Also if possible provide a reference for all of the diagnostic criteria in diagnosing AVN. 

Line78-79.   You may want to added a line describing the MCD score for the DHI.  This might be in important if you are discussing change in scores as an outcome to your study. 

Page 2 Line 87  Capitalize C in EyeSeeCam

Line 93.  Were vertical HIT also 150-250degrees/ sec.  EyeSeeCam allows for accepting lower velocities as low as 100 degree/sec for vertical head velocities.  If so, please clarify.

Page 3 Lines 102:  For the EyeSeeCam  there are different calculation.  Please be clear if the regression analysis for VOR gain and then asymmetry was used.  EyeSeeCam also reports instantaneous median VOR at 40, 60 and 80 ms.  The authors should be clear which calculations were used.

This is important because various vHIT manufactures calculate VOR gain different.

Line 104  Clearly define GA and Gc and Gi in the equation.

Paragraph starting on 108.  This description may be confusing to reader if unfamiliar with the testing planes.  Could the authors describe this with a figure (s)?

Line 113.  Clarify that Pathological saccades was considered “to be present in a patient” when…  As written, it is vague and made the results presentation confusing.  It appears that you counted the frequency that pathological saccades were present in each patient and calculated percent of patients with pathological saccades.  A figure that present overt and covert saccades with accepted amplitudes to define a pathological saccades may be of benefit. 

Statistical analysis

Line: 118-19  The authors should  replace word “referred” to reported or counted? 

Line 121:  These are parametric test.  Does the authors mean that looked and continuous variables in respect to the frequency of the categorical data?   It may just be better to describe with the categorical data was.  E.g.  presences of pathological saccades?

Results: 

Table 1 The data in the tables appear understandable, but title explanation does not match content.  The authors describe the canal plane combinations,  But use terms Affect HC and Health HC in the table.   Did the authors mean to define affected HC and the affected Horizontal canal plane and Health HC as the unaffected or contralateral horizontal canal plane?     Please rewrite description or use a figure (as noted in paragraph on line 108 above) to clarify.

Also for Controls.  How did the authors decide affected and Health canals planes.   Used in the tables as “Controls”.  This should be stated in methods.

Discussion. 

Page 5 staring on line 195.  Th authors compare results to recovery of VOR function from other studies.

Did the authors consider the medical management uses of anti-viral medication and vestibular rehabilitation may play a role in recovery.   Were patients in other studies treated the same or differently than those in your study.

Author Response

We thank the reviewer for his criticisms in terms of improving our article.

In Abstract  Page 1 line 18 Time for follow should be clearly stated. 

We have added "ranged from 6 to 30 months".

Introduction:  page 2 line 50.  Size of the “slow phase eye movement response.”  Eliminate term corrective.  That would be associated with the corrective saccades not the VOR.

We have correctly eliminated the word "corrective".

Materials=Methods

Line 63-64.  I would change location of word consecutively.  “The patients were admitted consecutively after diagnosed with AVN (no need to respell out) according to the following criteria:….”

As you suggested, we have changed location of word consecutively.

Line 67:  Please reference the Unterberger test or describe.   Also if possible provide a reference for all of the diagnostic criteria in diagnosing AVN. 

We have described the Unterberger test. We provided two references for all of the diagnostic criteria in diagnosing AVN.

Line78-79.   You may want to added a line describing the MCD score for the DHI.  This might be in important if you are discussing change in scores as an outcome to your study. 

In Material-Methods, last paragraph, last line, we have added the sentence "Pre- and …perceived handicap".

Page 2 Line 87  Capitalize C in EyeSeeCam

We have corrected it.

Line 93.  Were vertical HIT also 150-250degrees/ sec.  EyeSeeCam allows for accepting lower velocities as low as 100 degree/sec for vertical head velocities.  If so, please clarify.

We totally agree with you, the horizontal head velocities were higher (150–250o/s) than the vertical head velocities (50-150o/s). We added this sentence in vHIT paragraph.

Page 3 Lines 102:  For the EyeSeeCam  there are different calculation.  Please be clear if the regression analysis for VOR gain and then asymmetry was used.  EyeSeeCam also reports instantaneous median VOR at 40, 60 and 80 ms.  The authors should be clear which calculations were used.

This is important because various vHIT manufactures calculate VOR gain different.

We added the sentence "The EyeSeeCam system….gain values)." We used the instantaneous median VOR at 60 ms for measuring the horizontal gain.

Line 104  Clearly define GA and Gc and Gi in the equation.

As you suggested, we now clearly defined the GA, Gc and Gi in the equation; moreover, we have changed the position of this paragraph just below, to describe first the canal planes (affected – unaffected side).

Paragraph starting on 108.  This description may be confusing to reader if unfamiliar with the testing planes.  Could the authors describe this with a figure (s)?

We have added the Figure 1 in order to better clarify the testing planes.

Line 113.  Clarify that Pathological saccades was considered “to be present in a patient” when…  As written, it is vague and made the results presentation confusing.  It appears that you counted the frequency that pathological saccades were present in each patient and calculated percent of patients with pathological saccades.  A figure that present overt and covert saccades with accepted amplitudes to define a pathological saccades may be of benefit. 

For this, we have added the sentence "In our study, …calculated." to make clearer the definition of pathological saccades and the results. The figure 2 was also added with the accepted amplitudes of covert and overt saccades.

Statistical analysis

Line: 118-19  The authors should  replace word “referred” to reported or counted? 

We have replaced the word "referred" with the word "reported".

Line 121:  These are parametric test.  Does the authors mean that looked and continuous variables in respect to the frequency of the categorical data?   It may just be better to describe with the categorical data was.  E.g.  presences of pathological saccades?

Parametric tests were selected as the continuous variables analyzed followed a normal distribution. Respectively, the categorical variables, such as presence of pathological saccades, were presented in the form of frequencies and relative frequencies.

Results: 

Table 1 The data in the tables appear understandable, but title explanation does not match content.  The authors describe the canal plane combinations,  But use terms Affect HC and Health HC in the table.   Did the authors mean to define affected HC and the affected Horizontal canal plane and Health HC as the unaffected or contralateral horizontal canal plane?     Please rewrite description or use a figure (as noted in paragraph on line 108 above) to clarify.

We have replaced throughout the text and in Table 1 the term "healthy" with the term "unaffected" to make it clearer. In title explanation of Table 1, we have recommended to see the Figure 1 to better follow the head movements towards the affected and unaffected side, respectively, for each plane.

Also for Controls.  How did the authors decide affected and Health canals planes.   Used in the tables as “Controls”.  This should be stated in methods.

In Methods, the following sentence was added: "Depending on which ear was involved, the gain of the affected and the unaffected side was compared to the gain of the right and left ear of the healthy subjects (control group), respectively."

Discussion. 

Page 5 staring on line 195.  The authors compare results to recovery of VOR function from other studies.

Did the authors consider the medical management uses of anti-viral medication and vestibular rehabilitation may play a role in recovery.   Were patients in other studies treated the same or differently than those in your study.

It has been shown that neither corticosteroids [Fishman JM, Burgess C, Waddell A. Corticosteroids for the treatment of idiopathic acute vestibular dysfunction (vestibular neuritis). Cochrane Database Syst Rev. 2011] nor antiviral medication [Devaraja K. Is there enough evidence to refute the antiviral therapy in vestibular neuritis: A best evidence review. Indian J Otol 2018;24:135-8.] may play a role in recovery after acute vestibular neuritis. In our study, there were authors that used corticosteroids as sole treatment for acute vestibular neuritis [Allum & Honegger, Fu et al., Cerchiai et al], other authors did not report treatment in their studies [Magliulo et al., Riska et al., Navari et al.] and other stated only B12 as treatment [Buki et al.]; we have not found studies using antiviral medication for vestibular neuritis. So, we do not think that our medication for acute vestibular neuritis may influence the results compared to that of other studies.

Thank you very much for taking the time to comment, promoting our expression more clearly and getting improvement.

Best regards